# DRL-Sense: Deep Reinforcement Learning for Multi-Sense Word Representations

## Abstract

This paper proposes *DRL-Sense*—a multi-sense word representation learning model, to address the word sense ambiguity issue, where a sense selection module and a sense representation module are jointly learned in a reinforcement learning fashion. A novel reward passing procedure is proposed to enable joint training on the selection and representation modules. The modular design implements pure sense-level representation learning with linear time sense selection (decoding). We further develop a non-parametric learning algorithm and a sense exploration mechanism for better flexibility and robustness. The experiments on benchmark data show that the proposed approach achieves the state-of-the-art performance on contextual word similarities and comparable performance with Google's `word2vec` while using much less training data.

## 1 Introduction

Recently, deep learning methodologies have dominated several research areas in natural language processing (NLP), such as machine translation, language understanding, and dialogue systems. However, most of applications usually utilize word-level embeddings to obtain semantics. Considering that natural language is highly ambiguous, the standard word embeddings may suffer from polysemy issues. Neelakantan et al. (2014) pointed out that, due to triangle inequality in vector space, if one word embedding has two different senses, the sum of the distance between the word embedding and its synonymous embedding in each sense would be larger or equal to the distance between these synonymous embeddings[1]. Because this inaccurate distance measurement may degrade downstream NLP tasks, multi-sense word representations are proposed to address the ambiguity issue in a single word embedding scenario (Reisinger and Mooney, 2010; Huang et al., 2012)

This paper proposes *DRL-Sense*—a novel reinforcement learning based framework for learning multi-sense representations, which is composed by two main modules: *a sense selection module* for inferring the most probable sense for a word given its context, and *a sense representation module* for representing word senses in a continuous space. Our modular design implements pure sense-level representation learning while maintaining linear time sense selection. The proposed model is optimized by reinforcement learning and incorporates a non-parametric algorithms and a sense exploration mechanism without changing the network architecture[2]. Our contributions are four-fold:

- We are among the first to study reinforcement learning for sense embedding learning, taking account of the Markov property in sense selection given local contexts.
- *DRL-Sense* is the first system that achieves pure sense-level representation learning with linear time complexity on sense selection.
- We develop non-parametric learning and sense exploration mechanisms for a general neural sense selection module to achieve better flexibility and robustness.
- Our experimental results show the state-of-the-art performance on contextual word similarities and comparable performance with `word2vec` (Mikolov et al., 2013b) using only 1/100 size of training data.

---

[1] $d(\text{bank}, \text{finance}) + d(\text{bank}, \text{coast}) \geq d(\text{finance}, \text{coast})$

[2] The code will be released after the paper gets accepted.

## 2 Related Work

There are three dominant types of approaches for learning multi-sense word representations in the literature: 1) clustering methods, 2) probabilistic modeling methods, and 3) lexical ontology based methods. Our reinforcement learning based approach can be loosely connected to clustering methods and probabilistic modeling methods.

Reisinger and Mooney (2010) is the first work to propose multi-sense word representations on vector space based on clustering techniques. With the power of deep learning, some work exploit neural networks to learn embeddings with sense-annotated corpus based on clustering (Huang et al., 2012; Neelakantan et al., 2014). Chen et al. (2014) replaced the clustering procedure with a word sense disambiguation model based on Word-Net (Miller, 1995). Kågebäck et al. (2015) further exploited a weighting mechanism on context in the clustering procedure and evaluated their system on word sense induction. Vu and Parker (2016) proposed an iterative process on the two-stage clustering-embedding learning framework. Moreover, Guo et al. (2014) leveraged bilingual resources to perform clustering. However, most of the above approaches separates the clustering procedure and the representation learning procedure, which may suffer from the error propagation issue. Instead, our reinforcement learning model utilizes a reward signal to propagate the statistical information from sense representations to optimize sense selection.

In order to make the model more flexible, probabilistic modeling methods are usually utilized for learning multi-sense emnbeddings, where Tian et al. (2014) and Jauhar et al. (2015) conducted probabilistic modeling with EM training. Li and Jurafsky (2015) exploited the Chinese Restaurant Process and demonstrates efficacy of multi-sense word representations on several downstream NLP tasks. Furthermore, Bartunov et al. (2016) developed a non-parametric Bayesian extension on the skip-gram model (Mikolov et al., 2013b). However, all above methods fail to maintain pure sense-level representation learning due to the complicated computation in their EM algorithms (Bartunov et al., 2016), where sense-level embeddings are trained with word-level embeddings.

Recently, Qiu et al. (2016) proposed an EM algorithm to learn representations without word-level embeddings, where the computational cost is high when selecting the sense identity sequence because the time for searching all sense combination within a context window is exponential. In contrast, by exploiting embeddings in the sense selection module, our model performs linear time sense selection, while maintaining a pure sense-level representation learning module. Another difference between the prior work and ours is that, Qiu et al. (2016) used WordNet (Miller, 1995) to obtain the number of senses for words, and our solution supports non-parametric learning for automatically deciding the sense number of each word.

Furthermore, most related work either directly exploited sense representations (Neelakantan et al., 2014; Qiu et al., 2016) or combined with online expectation sense counts (Li and Jurafsky, 2015; Jauhar et al., 2015) to select senses. The proposed approach incorporates a sense selection module, which can be learned together with the representation learning module through reinforcement learning.

Unlike a lot of relevant work that requires additional resources such as the lexical ontology (Rothe and Schütze, 2015; Jauhar et al., 2015; Chen et al., 2015; Iacobacci et al., 2015) or bilingual data (Guo et al., 2014; Ettinger et al., 2016; Šuster et al., 2016), which may not be available in the target task, our model can be trained on the corpus only. Also, some prior work proposed to learn topical embeddings and word embeddings jointly in order to consider the contexts (Liu et al., 2015a,b), whereas this paper focuses on learning multi-sense word embeddings.

## 3 Proposed Approach: DRL-Sense

This work proposes a framework to learn two key modules for multi-sense word representations: *a sense selection module* and *a sense representation module*. The sense selection module decides which sense to use given a text context, whereas the sense representation module learns meaningful representation for sense by their statistical characteristics.

Considering that the sense representation module requires the sense identity of each word from the sense selection module, and the sense selection module may also benefit from the semantics carried by sense representations, these two modules should be tangled. Hence, a naive two-stage algorithm or two separate learning algorithms proposed by prior work are not optimal. We propose

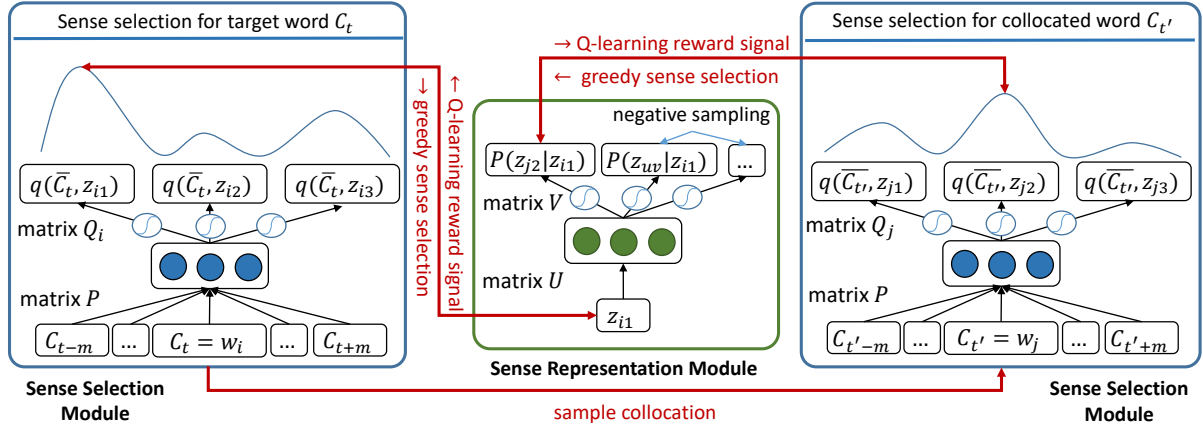

Figure 1: The *RRL-Sense* architecture illustration. Two shared-weight sense selection modules surround a sense representation module. The sense selection module infers the target word sense given its context. The selected senses are passed to the sense representation module for learning sense embeddings. Then, the sense embedding collocation is passed back to the sense selection modules for reinforcement learning.

to learn two modules in a reinforcement learning manner by introducing a novel reward passing procedure to enable joint training. In addition, a non-parametric learning algorithm is incorporated to enable automatic sense induction, where the number of senses for each word is not predefined but automatically learned for practical usage. Also, simple mechanisms are applied to allow the sense exploration while incorporating the sense selection prior.

## 3.1 Model Architecture

Our model architecture is illustrated in Figure 1 and detailed below.

### 3.1.1 Sense Selection Module

The sense selection module decides the sense for a word given its context. Assuming that a word sense is determined by the local context, we use the Markov property to formulate the sense selection module as a Markov Decision Process (MDP) to infer the most probable sense based on its local context. Given a corpus $C$ and a context window $m$, we extract a local context $\bar{C}_t = \{C_{t-m}, ..., C_{t-1}, C_{t+1}, ..., C_{t+m}\}$ as a state $S_t$, and the selection of a sense $z_{ik} \in Z_i$ for a word $w_i = C_t$ given context $\bar{C}_t$ as an action $A_{ik}$ given the state $S_t$, where $Z_i$ is the set of senses of word $w_i$. However, this formulation lacks a central element of MDP: a reward signal for measuring the fitness of the selected sense $z_{ik}$. Therefore, if we have meaningful sense representations containing statistical estimation, we can use the estimation as a surrogate reward to simulate the reward signal in the sense selection module.

### 3.1.2 Sense Representation Module

A successful sense selection module can be applied to each word in a corpus for obtaining its sense identity. Various techniques about word embeddings can be directly employed after mapping all words in a corpus to its sense identity. The typical method is to formulate the sense selection problem as a maximum likelihood estimation (MLE) problem for the collocation likelihood. In this paper, we use the skip-gram formulation (Mikolov et al., 2013b), because it is more light-weight and thus requires less training time: only two sense selections are required for stochastic training, while GloVe (Pennington et al., 2014) and continuous bag-of-words (CBOW) (Mikolov et al., 2013a) require the whole corpus and the whole context window for sense selection respectively.

Specifically, we first create input sense embedding matrix $U$ and collocation estimation matrix $V$ as the representations to be learned. Given a word $w_i$ and collocated word $w_j$ with corresponding local contexts, we map them to their sense identities as $z_{ik}$ and $z_{jl}$ by the sense selection module, and maximize the sense collocation log likelihood $\log Co(\cdot)$ using a negative sampling approximation (Mikolov et al., 2013b),

$$\log Co(z_{ik}, z_{jl} \mid U, V) = \log \sigma(U_{z_{ik}}^T V_{z_{jl}}) \quad (1)$$
$$+ \sum_{v=1}^{M} \mathbb{E}_{z_{uv} \sim p_{neg}(z)} [\log \sigma(-U_{z_{ik}}^T V_{z_{uv}})],$$

where $p_{neg}(z)$ is the distribution over all senses for negative samples.

## 3.2 Reinforcement Learning

With the above two modules, we treat the sense collocation likelihood as a surrogate reward signal for the sense selection module, and then cast the whole formulation as a reinforcement learning task: An optimal policy $\pi_\theta$ with parameters $\theta$ should select the sense that leads to an optimal collocation likelihood in (1). In order to estimate the sense collocation likelihood, two senses, $z_{ik}$ and $z_{jl}$, should be determined, which leads a two-step MDP: one for the target word $w_i$ and the other for the collocated word $w_j$ within a context window. A policy gradient algorithm (Sutton et al., 1999) can formulate the policies as decision probability distributions $\pi_\theta^i(z_{ik}|\bar{C}_t)$ and $\pi_\theta^j(z_{jl}|\bar{C}_{t'})$ for words $w_i, w_j$ given their contexts $\bar{C}_t, \bar{C}_{t'}$. Hence, a joint formulation can be written as follows,

$$\max_{U,V,\theta} \mathbb{E}_{k\sim\pi_\theta^i(\cdot)}[\mathbb{E}_{l\sim\pi_\theta^j(\cdot)}[\log Co(z_{ik}, z_{jl} \mid U, V)]].$$
(2)

The neat formulation is differentiable and supports stochastic optimization using doubly stochastic gradient (Lei et al., 2016). However, there are two major drawbacks of this formulation. First, it requires a policy $\pi_\theta$ to fit a valid probability distribution, and this constraint underestimates the probability of senses that have not been selected yet during optimization. Second, if using stochastic gradient ascent (detailed in Appendix A) to optimize equation (2), it would actually lower the probability estimation for the selected sense because $\log Co(\cdot) \leq 0$, which is not desirable if the model accurately select the right sense.

### 3.2.1 Q-Learning Formulation

To address the above issue, a Q-learning algorithm (Mnih et al., 2013) with a novel reward passing procedure is applied to this task, because it optimizes the selected action (sense) independently to the other actions. Also, considering that the Q-learning algorithm is not constrained by a *joint-probabilistic* formulation, we reduce the range of the reward function to facilitate the training process by replacing the collocation log likelihood $\log Co \in (-\inf, 0]$ with (its monotonous) likelihood $Co \in [0, 1]$ as the reward function (Mnih et al., 2013).

Our model utilizes a CBOW architecture to model the fitness for a sense $z_{ik}$ of a word $w_i$ under a local context $\bar{C}_t$. Specifically, given a *word* embedding matrix $P$, the local context can be modeled as the summation of word embeddings from

---

**Algorithm 1:** DRL-Sense Training Algorithm

> **for** $w_i = C_t \in C$ **do**
> > sample $w_j = C_{t'}(0 < |t' - t| \leq m)$;
> > $k = \arg\max_{k'} q(\bar{C}_t, z_{ik'})$;
> > $l = \arg\max_{l'} q(\bar{C}_{t'}, z_{jl'})$;
> > optimize $U, V$ by (1) for the sense representation module;
> > optimize $P, Q$ by (4) for the sense selection module;

---

its context $\bar{C}_t$. The fitness representation of the selected sense can be formulated with a 3-mode tensor $Q$, whose dimensions denote words, senses, and latent variables. Then we model the Q-value, $q(\bar{C}_t, z_{ik})$, for selecting the sense $z_{ik}$ under the context $\bar{C}_t$ as

$$q(\bar{C}_t, z_{ik}) = \sigma(Q_{ik}^T \sum_{j\in\bar{C}_t} P_j), \qquad (3)$$

where $\sigma$ is a sigmoid function to form a valid probabilistic estimation.

### 3.2.2 DRL-Sense Training

To jointly train sense selection and sense representation modules, we first sample a collocated word pair $w_i, w_j$ with respective contexts $\bar{C}_t, \bar{C}_{t'}$, and use the estimated Q-value to select the most probable senses $z_{ik}, z_{jl}$. The selected senses are passed to the sense representation module to optimize the sense collocation likelihood. Afterwards, the estimated collocation likelihood is passed back as a reward signal to optimize the sense selection module. The loss function is defined as cross-entropy $H(\cdot)$ due to the probability distribution.

$$\min_{P,Q} \; H(Co(z_{ik}, z_{jl} \mid U, V), q(\bar{C}_t, z_{ik}))$$
$$+ H(Co(z_{ik}, z_{jl} \mid U, V), q(\bar{C}_{t'}, z_{jl})). \quad (4)$$

Algorithm 1 shows the proposed DRL-Sense model training procedure. There are two major contributions in our modular design. First, efficient sense selection with word embeddings and pure sense representation learning are simultaneously achieved. Second, reinforcement learning allows both modules to be jointly trained.

## 3.3 Non-Parametric Learning

In order to automatically discover the sense number for each word, we incorporate word sense induction into our model (Song et al., 2016). Instead

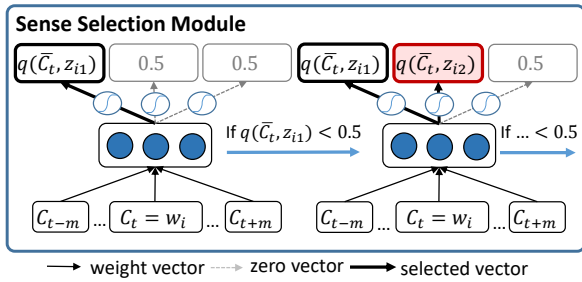

Figure 2: The proposed non-parametric algorithm with initialization. The red block indicates the newly discovered sense $z_{i2}$ for the target word $w_i$ based on the algorithm.

of non-parametric clustering (Neelakantan et al., 2014) or the stochastic process (Li and Jurafsky, 2015) applied by prior work, this paper proposes a novel mechanism in a general neural model to conduct non-parametric sense discovery.

Because the formulated Q-value is a probabilistic estimation, the model allows the selected word to create a new sense when all Q-values of its existing senses are lower than $0.5$. Therefore, the prediction layer is initialized to $0$ so that all Q-values remain $\sigma(0) = 0.5$ until having been selected due to independent updates in Q-learning. Figure 2 illustrates the sense discovery procedure, where the model expands a word's sense set when its context has all Q-values of existing senses less than $0.5$.

### 3.4 Word Sense Exploration

Due to high ambiguity in natural language, a greedy sense selection strategy may not work well in the early training stage, because the sense selection module does not learn well. Such a drawback also exists in the literature using clustering algorithms (Neelakantan et al., 2014; Kågebäck et al., 2015) and hard-EM algorithms (Qiu et al., 2016; Jauhar et al., 2015). This issue is known as exploration-exploitation trade-off (Sutton and Barto, 1998). Li and Jurafsky (2015) proposed to sample on a Chinese Restaurant Process to introduce uncertainty. In our neural network architecture for computing Q-values, we perform dropout (Srivastava et al., 2014) in the hidden layer $\sum_{j \in \bar{C}_t} P_j$ for our sense selection module to introduce uncertainty.

### 3.5 Sense Selection Prior

Incorporating priors in sense selection is a key element for probabilistic models (Li and Juraf-

sky, 2015; Jauhar et al., 2015), most previous work used online selection counts of each sense as its prior. As the key utility of the prior is to incorporate the sense selection preference for a word $w_i$ (disregarding its context $\{C_{t-m}, ..., C_{t-1}, C_{t+1}, ..., C_{t+m}\}$) in its sense selection process, we achieve similar prior utility by adding the target word $w_i$ into the input of the sense selection module as $\bar{C}_t = \{C_{t-m}, ..., C_{t+m}\}$ such that the Q-values indicate the sense preference of the target word.

## 4 Experiments

We evaluate our proposed DRL-Sense model and compare with other multi-sense word representation models for both quantitative and qualitative experiments.

### 4.1 Experimental Setup and Configuration

Our model is trained on the April 2010 Wikipedia dump (Shaoul and Westbury, 2010), which contains approximately 1 billion tokens. For fair comparison, we adopt the same vocabulary set as Huang et al. (2012) and Neelakantan et al. (2014). For preprocessing, we convert all words to lower case, apply the Stanford tokenizer and the Stanford sentence tokenizer (Manning et al., 2014), and remove all sentences with less than 10 tokens.

In the experiments, the context window size is set to 6 ($|\bar{C}_t| = 13$). Subsampling technique introduced by word2vec (Mikolov et al., 2013b) is applied to accelerate the training process. The (upper bound of) number of senses per word in $Q$ is set to 3 for fair comparison with prior work (Neelakantan et al., 2014). The learning rate is set to 0.025. Dropout is annealed from 0.5 to 0.0 within the first epoch. The embedding dimension is 300. We initialize $Q$ and $V$ as zeros, and $P$ and $U$ as the uniform distribution $[-\sqrt{1/100}, \sqrt{1/100}]$ such that each embedding has unit length in expectation (Lei et al., 2015). For the negative sampling distribution in (1), we use $1/3$ unigram of a word for each of its sense to compute the negative sampling probability using the 3/4rd power trick in Mikolov et al. (2013b). Our model uses 5 negative senses for negative sampling in (1).

In optimization, we conduct mini-batch training with 2048 batch size using the following procedure: 1) select senses in the batch; 2) optimize $U, V$ using stochastic training within the batch for efficiency; 3) optimize $P, Q$ using mini-

| Method | MaxSimC | AvgSimC |
|---|---|---|
| Huang et al. (2012) | 26.1 | 65.7 |
| Neelakantan et al. (2014) | 60.1 | 68.6 |
| Tian et al. (2014) | 63.6 | 65.4 |
| Li and Jurafsky (2015) | 66.4 | - |
| Bartunov et al. (2016) | 53.8 | 61.2 |
| Qiu et al. (2016) | 64.9 | 66.1 |
| Proposed: DRL-Sense | **66.6** | 65.2 |
| - Non-Parametric | 65.1 | 67.0 |
| - Sense Exploration | 66.2 | 64.5 |
| - Sense Selection Prior | 66.3 | 66.0 |

Table 1: Spearman's rank correlation $\rho$ x100 on the SCWS dataset.

batch training for robustness. To further stabilize the reward signal for the sense selection module, we only use the collocated sense $\sigma(U_{z_{ik}}^T, V_{z_{jl}})$ in (1) to approximate the collocation likelihood $Co(z_{ik}, z_{jl} \mid U, V)$ for Q-learning in (4).

## 4.2 Experiment 1: Contextual Word Similarity

To evaluate the quality of sense embeddings, we compute the similarity score between each word pair given their respective local contexts and compare with the human-judged score using Stanford's Contextual Word Similarities (SCWS) data (Huang et al., 2012). Specifically, given a list of word pairs with corresponding contexts, $S = \{(w_i, \bar{C}_t, w_j, \bar{C}_{t'})\}$, we calculate the Spearman's rank correlation $\rho$ between human-judged similarity and model similarity estimations. Two major contextual similarity estimations are introduced by Reisinger and Mooney (2010): AvgSimC and MaxSimC (also referred to LocalSim in Neelakantan et al. (2014)). AvgSimC is a *soft* measurement that addresses the contextual information with a probability estimation:

$$\text{AvgSimC}(w_i, \bar{C}_t, w_j, \bar{C}_{t'}) =$$
$$\sum_{k=1}^{Z_i} \sum_{l=1}^{Z_j} p(z_{ik}|\bar{C}_t)p(z_{jl}|\bar{C}_{t'})d(z_{ik}, z_{jl}),$$

which weights the similarity measurement of each sense $z_{ik}$ and $z_{jl}$ by their probability estimations. On the other hand, MaxSimC is a *hard* measurement that only evaluates on the most probable senses:

$$\text{MaxSimC}(w_i, \bar{C}_t, w_j, \bar{C}_{t'}) = d(z_{ik}, z_{jl}),$$
$$z_{ik} = \arg\max_{k'} p(z_{ik'}|\bar{C}_t), z_{jl} = \arg\max_{l'} p(z_{jl'}|\bar{C}_{t'}).$$

The baselines for comparison include classic clustering methods (Huang et al., 2012; Neelakantan et al., 2014), EM algorithms (Tian et al., 2014; Qiu et al., 2016; Bartunov et al., 2016), and Chinese Restaurant Process (Li and Jurafsky, 2015), where all approaches are trained on the same corpus except Li and Jurafsky (2015)[3] and Qiu et al. (2016) used more recent Wikipedia dumps. The embedding sizes of all baselines are 300, except 50 in Huang et al. (2012). In our method, the probability assignment on each sense $p(z_{ik}|\bar{C}_t)$ is calculated by replacing the sigmoid layer with a softmax layer in our sense selection module as:

$$p(z_{ik} \mid \bar{C}_t) = \frac{\exp(Q_{ik}^T \sum_{j \in \bar{C}_t} P_j)}{\sum_{k' \in Z_i} \exp(Q_{ik'}^T \sum_{j \in \bar{C}_t} P_j)}$$

Table 1 shows the experimental results. An ablation test on each component in our system is shown in the lower part of the table, where each learning strategy can effectively improve the DRL-Sense performance[4].

### 4.2.1 Embedding Quality Analysis

Our DRL-Sense model achieves the state-of-the-art performance on MaxSimC, demonstrating superior quality on independent sense embeddings. In real world applications, using multiple sense vectors for a word simultaneously may bring additionally computational overhead over conventional single word embedding scenarios, and also change the existing neural network architecture for word embeddings. In contrast, by simply replacing the most probable sense identity with each word identity (as in MaxSimC), the computation framework and cost for downstream NLP tasks remains the same as conventional word embedding approaches. Rothe and Schütze (2015) reported that the traditional single word representation model word2vec (Mikolov et al., 2013b), released by Google and trained on 100 billion tokens Google's internal dataset, can achieved 66.6% similarity correlation on the SCWS dataset. Table 1 shows that our proposed DRL-Sense achieves comparable performance as word2vec using only 1/100 size of Google's internal data, while using multi-sense representations in the single embedding scenario (MaxSimC).

---

[3] We use Li and Jurafsky (2015)'s result on 1 billion tokens for fair comparison.

[4] More detail is available in Appendix B.

| Method | ESL-50 | RD-300 | TOEFL-80 |
|---|---|---|---|
| *(1) Conventional Word Embedding* | | | |
| GC | 47.73 | 45.07 | 60.87 |
| SG | 52.08 | 55.66 | 66.67 |
| *(2) Word Sense Disambiguation* | | | |
| IMS+SG | 41.67 | 53.77 | 66.67 |
| *(3) Direct Learning from Corpus* | | | |
| EM | 27.08 | 33.96 | 40.00 |
| NP-MSSG | 45.24 | 50.00 | **81.16** |
| **DRL-Sense** | **54.76** | **60.71** | 72.46 |
| *(4) Retrofitting with WordNet* | | | |
| Retro-GC | 63.64 | 66.20 | 71.01 |
| Retro-SG | 56.25 | 65.09 | 73.33 |

Table 2: Accuracy on synonym selection

### 4.2.2 Probability Assignment Analysis

To interpret the performance between MaxSimC and AvgSimC, we claim that an ideal model should have MaxSimC≥AvgSimC in Table 1. The reason is that, any sense selection model seeks a function $f(z_{ik} \mid \bar{C}_t)$ to evaluate the fitness of each sense $z_{ik}$ under a context $\bar{C}_t$, such as the cluster similarity in clustering methods, the likelihood or vanilla probability estimation in probabilistic models, and the Q-value in our model. Hence, to achieve the optimal expected fitness $\mathbb{E}[f(z_{ik} \mid \bar{C}_t)]$, the optimal probability assignment on senses is only selecting one and the optimal sense:

$$p_{opt}(z_{ik} \mid \bar{C}_t) = \begin{cases} 1 & \text{if } k = \arg\max_{k'} f(z_{ik'} \mid \bar{C}_t), \\ 0 & \text{otherwise.} \end{cases}$$

Note that the above probability assignment makes AvgSimC=MaxSimC. Because the expected fitness $\mathbb{E}[f(z_{ik} \mid \bar{C}_t)]$ over any probability distribution would be no greater than the fitness value over the above distribution, the similar phenomenon (MaxSimC≥AvgSimC) can be observed in the robust fitness estimation (cluster similarity/likelihood/Q-value). Therefore, from Table 1, our DRL-Sense model demonstrates not only superior performance under a single sense embedding scenario but also reasonable and robust sense selection.

### 4.3 Experiment 2: Synonym Selection

We further evaluate our model on synonym selection using multi-sense word representations (Jauhar et al., 2015). Three standard synonym selection datasets, ESL-50 (Turney, 2001), RD-300 (Jarmasz and Szpakowicz, 2004), and

TOEFL-80 (Landauer and Dumais, 1997), are performed. In the datasets, each question consists of a question word $w_Q$ and four answer candidates $\{w_A, w_B, w_C, w_D\}$, and the goal is to select the most semantically synonymous choice among the four candidates. For example, in the TOEFL-80 dataset, a question shows {(Q) enormously, (a) appropriately, (b) uniquely, (c) tremendously, (d) decidedly}, and the answer is (c).

In the experiments, our model selects the synonym of the question word $w_Q$ by the collocation likelihood as a proxy of their semantic similarity. That is, for pairs of words $(w_Q, w_T), T \in \{A, B, C, D\}$, we select the pair with the maximum bi-directional collocation likelihood,

$$\begin{aligned} &\text{Sim}(w_Q, w_T) \\ &= \max_{j \in Z_Q, l \in Z_T} \max(Co(z_{Qj}, z_{Tl}), Co(z_{Tl}, z_{Qj})) \\ &= \max_{j \in Z_Q, l \in Z_T} \max(\sigma(U_{Qj}^T V_{Tl}), \sigma(U_{Tl}^T V_{Qj})). \end{aligned}$$

Our model is compared with following baselines: (1) conventional word embedding: global context vectors (GC) (Huang et al., 2012) and skip-gram (SG) (Mikolov et al., 2013b); (2) applying supervised word sense disambiguation (using the IMS system (Zhong and Ng, 2010)) and then applying SG on disambiguated corpus: IMS+SG; (3) direct training multi-sense word representations on corpus: EM algorithm (Jauhar et al., 2015), non-parametric multi-sense skip-gram (NP-MSSG) (Neelakantan et al., 2014) and our DRL-Sense; (4) retrofitting existing word embeddings to sense embeddings using WordNet: retrofitting GC (Retro-GC) and retrofitting SG (Retro-SG) (Jauhar et al., 2015). Note that we do not compare with the methods in (4) on Table 1, because sense representations are generated without a sense selector, and thus cannot be evaluated using MaxSimC and AvgSimC. Also, the methods in (4) can be treated as an upperbound of the methods in (3) due to the usage of additional supervised information from WordNet.

The experimental results are shown on Table 2. Except for retrofitting methods and NP-MSSG in TOEFL-80, our DRL-Sense model significantly outperforms all baselines. In addition, our model also performs better than Retro-GC for TOEFL-80, although Retro-GC uses the supervised knowledge from WordNet, showing that our model obtains good sense embeddings by learning with the robust sense selector. Moreover, our method out-

| Sense | k-NN Senses |
|---|---|
| mercury-1 | mercury comet apollo comets wnba |
| mercury-2 | mercury sulfide beryllium zirconium |
| apple-1 | apple blackberry macintosh ibm novell |
| apple-2 | avocado apple apricot blackberry prunus |
| lucky-1 | lucky gotti co-starring gravano starred |
| lucky-2 | beyonce singin beenie baby choo lil |
| lucky-3 | hilfiger starbucks t-bone puff armani |

Table 3: Sense representations with neighboring words using the collocation likelihood.

performs the EM method and NP-MSSG (Neelakantan et al., 2014), despite the highest score of AvgSimC on Table 1, echoing the superior quality of our sense vectors as suggested in the MaxSimC measurement.

### 4.4 Qualitative Analysis

We qualitatively evaluate non-parametric learning and sense representation learning performance.

#### 4.4.1 Non-Parametric Learning Effectiveness

The largest challenge of multi-sense representation learning is that most words can be represented by a single embedding (non-polysemous or a single embedding can model multiple senses well). Hence, the feasible solution is to retain most words a single sense embedding, while leaving polysemous words multiple sense embeddings. Huang et al. (2012) and Neelakantan et al. (2014) only selected approximately 6,000 and 30,000 words from a vocabulary with size ≈ 100,000 for multi-sense representation learning. Qiu et al. (2016) utilized a coarse version (Navigli et al., 2007) of WordNet inventory (Miller, 1995) to determine the sense number for each word before training. The sense selection module in our learned DRL-Sense produces 71,194 words with a selected sense, 22,271 words with 2 senses, and 5,683 words with 3 senses based on the training data. Without any multi-sense determination procedure prior to training, our method automatically determines only about 30,000 polysemous words, where the number is close to Neelakantan et al. (2014)'s setting, demonstrating the effectiveness of our non-parametric learning algorithm.

#### 4.4.2 Representation Learning Effectiveness

To evaluate the quality of sense embeddings, we show the k-nearest neighbors (k-NN) for each sense. For example, given the first sense of the word "apple", denoted as "apple-1", we select

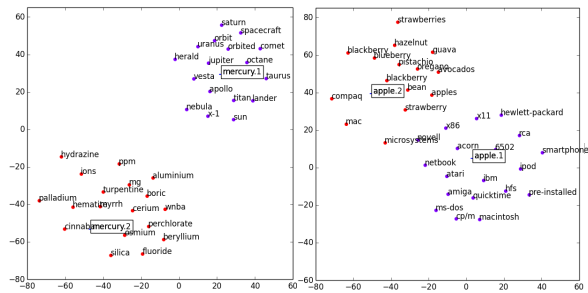

Figure 3: The 2-D visualization using t-SNE (Maaten and Hinton, 2008) for the words "mercury" and "apple" on multi-sense word representations with $L_2$ distance neighbors.

$k$ senses with the highest collocation likelihood from the sense representation module and list in Table 3[5]. The learned sense embeddings of the words "mercury" and "apple" clearly correspond to different senses: "planet" (mercury-1), "metal element" (mercury-2), "computer" (apple-1), and "fruit" (apple-2) as visualized in Figure 3. Instead of actual senses, Table 3 also shows that our model is able to automatically detect a fine-grained *aspect* division within a word "lucky" in an unsupervised manner. Specifically, it divides the semantics of "lucky" into three aspects: "starring" (lucky-1), "musician" (lucky-2), and "brands" (lucky-3), expanding the potential usage of the learned embeddings[6].

## 5 Conclusion

We propose *DRL-Sense*—a novel deep reinforcement learning framework to jointly learn a word sense selection module and a sense representation module. Our model implements non-parametric learning for word sense induction and exploration for word sense selection. The experiments show that our DRL-Sense model achieves the state-of-the-art performance for the benchmark contextual word similarity task and most of synonym selection datasets under the same setting. In the future, we plan to investigate reinforcement learning methods to incorporate multi-sense word representations for downstream NLP tasks.

---

[5] For clarity purpose, we omit the sense identity in the k-NN senses.

[6] Note that the sense induction procedure highly depends on the training corpus, so it may fail to discover all senses in an existing sense inventory.

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

## A  Doubly Stochastic Gradient

For clarity purpose, we abbreviate the collocation log likelihood $\log Co(ik, jl|U, V)$ as $\mathcal{L}(\cdot)$ in this section. To derive doubly stochastic gradient for equation (2), we first resolve the expectation form as:

$$J(U, V, \theta)$$
$$= \mathbb{E}_{k \sim \pi_\theta^i(\cdot)}[\mathbb{E}_{l \sim \pi_\theta^j(\cdot)}[\mathcal{L}(\cdot)]]$$
$$= \sum_k \sum_l \pi_\theta^i(z_{ik}|\bar{C}_t)\pi_\theta^j(z_{jl}|\bar{C}_{t'})\mathcal{L}(\cdot).$$

The gradient of $J$ with respect to $\Theta$ should be:

$$\frac{\partial J(U, V, \theta)}{\partial \Theta}$$
$$= \frac{\partial}{\partial \Theta} \sum_k \sum_l \pi_\theta^i(z_{ik}|\bar{C}_t)\pi_\theta^j(z_{jl}|\bar{C}_{t'})\mathcal{L}(\cdot)$$
$$= \sum_k \sum_l \pi_\theta^i(z_{ik}|\bar{C}_t)\mathcal{L}(\cdot)\frac{\partial \pi_\theta^j(z_{jl}|\bar{C}_{t'})}{\partial \Theta}$$
$$+ \sum_k \sum_l \pi_\theta^j(z_{jl}|\bar{C}_{t'})\mathcal{L}(\cdot)\frac{\partial \pi_\theta^i(z_{ik}|\bar{C}_t)}{\partial \Theta}$$
$$= \mathbb{E}_{k \sim \pi_\theta^i(\cdot)}[\sum_l \mathcal{L}(\cdot)\frac{\partial \log \pi_\theta^j(z_{jl}|\bar{C}_{t'})}{\partial \Theta}\pi_\theta^j(z_{jl}|\bar{C}_{t'})]$$
$$+ \mathbb{E}_{l \sim \pi_\theta^j(\cdot)}[\sum_k \mathcal{L}(\cdot)\frac{\partial \log \pi_\theta^i(z_{ik}|\bar{C}_t)}{\partial \Theta}\pi_\theta^i(z_{ik}|\bar{C}_t)]$$
$$= \mathbb{E}_{k \sim \pi_\theta^i(\cdot)}[\mathbb{E}_{l \sim \pi_\theta^j(\cdot)}[\mathcal{L}(\cdot)$$
$$\frac{\partial}{\partial \Theta}(\log \pi_\theta^i(z_{ik}|\bar{C}_t) + \log \pi_\theta^j(z_{jl}|\bar{C}_{t'}))]]$$

Accordingly, if we conduct typical stochastic gradient ascent training on $J(U, V, \Theta)$ with respect to $\Theta$ from samples $k$ and $l$ with a learning rate $\eta$, the update formula will be:

$$\Delta\Theta = \eta\mathcal{L}(\cdot)\frac{\partial}{\partial \Theta}(\log \pi_\theta^i(z_{ik}|\bar{C}_t) + \log \pi_\theta^j(z_{jl}|\bar{C}_{t'})).$$

However, because the collocation likelihood is a valid probability distribution, the collocation log likelihood should always be non-positive: $\mathcal{L}(\cdot) = \log Co(z_{ik}, z_{jl}|U, V) \leq 0$. Therefore, as long as the collocation log likelihood $\mathcal{L}$ is negative, the update formula is to minimize the likelihood of choosing $l$ and $k$, despite the fact that $l$ and $k$ may be good choices. On the other hand, if the log likelihood reaches 0, it also indicates that the likelihood reaches infinity and computational overflow on $U$ and $V$.

| Method | MaxSimC | AvgSimC |
|---|---|---|
| Proposed: DRL-Sense | **66.6** | 65.2 |
| - non-parametric | 65.1 | 67.0 |
| - sense exploration | 66.2 | 64.5 |
| - sense selection prior | 66.3 | 66.0 |
| 50D-300D | 66.0 | 65.8 |
| 50D-50D | 65.0 | 64.6 |

Table 4: Spearman's rank correlation $\rho$ x100 on the SCWS dataset for ablation experiments.

## B  Supplementary Experiments

In this section, we conduct ablation experiments by removing each components separately in our system to test the efficacy of proposed mechanisms. Specifically, we test the following components:

- Non-parametric: To remove the non-parametric learning mechanism in our model, we initialize the prediction layer $Q$ in the sense selection module from uniform distribution $[-\sqrt{1/100}, \sqrt{1/100}]$ as the word embedding $P$.

- Sense exploration: To remove the word sense exploration mechanism in our model, we turn off the dropout mechanism during training.

- Sense selection prior: To remove the sense selection prior mechanism in our model, we remove the target word $w_i$ in the input of the sense selection module as $\bar{C}_t = \{C_{t-m}, ..., C_{t-1}, C_{t+1}, ..., C_{t+m}\}$.

In addition, we also test the efficacy of specific module according to their complexity. We test the setting when the embedding dimension in the sense selection module is set to 50 (50D/300D), and the embedding dimensions in the both modules are set to 50 (50D/50D).

The experiment results are shown in Table 4. Consistent improvement in terms of MaxSimC from each component to the proposed DRL-Sense model can be observed. In addition, the improvement from increasing the complexity (embedding dimension) demonstrate the efficacy of both modules (50D-50D→50D-300D and 50D-300D→DRL-Sense).

