# Peer review of "DRL-Sense: Deep Reinforcement Learning for Multi-Sense Word Representations"

_ACL 2017 — decision unknown_

[Official Review · Reviewer 1 · rating 3 · confidence 4]
soundness 5 · originality 5 · clarity 4 · impact 3 · substance 3 · appropriateness 5 · meaningful comparison 3 · presentation format Poster

This paper outlines a method to learn sense embeddings from unannotated corpora
using a modular sense selection and representation process. The learning is
achieved by a message passing scheme between the two modules that is cast as a
reinforcement learning problem by the authors.

- Strengths:

The paper is generally well written, presents most of its ideas clearly and
makes apt comparisons to related work where required. The experiments are well
structured and the results are overall good, though not outstanding. However,
there are several problems with the paper that prevent me from endorsing it
completely.

- Weaknesses:

My main concern with the paper is the magnification of its central claims,
beyond their actual worth.

1) The authors use the term "deep" in their title and then several times in the
paper. But they use a skip-gram architecture (which is not deep). This is
misrepresentation.

2) Also reinforcement learning is one of the central claims of this paper.
However, to the best of my understanding, the motivation and implementation
lacks clarity. Section 3.2 tries to cast the task as a reinforcement learning
problem but goes on to say that there are 2 major drawbacks, due to which a
Q-learning algorithm is used. This algorithm does not relate to the originally
claimed policy.

Furthermore, it remains unclear how novel their modular approach is. Their work
seems to be very similar to EM learning approaches, where an optimal sense is
selected in the E step and an objective is optimized in the M step to yield
better sense representations. The authors do not properly distinguish their
approach, nor motivative why RL should be preferred over EM in the first place.

3) The authors make use of the term pure-sense representations multiple times,
and claim this as a central contribution of their paper. I am not sure what
this means, or why it is beneficial.

4) They claim linear-time sense selection in their model. Again, it is not
clear to me how this is the case. A highlighting of this fact in the relevant
part of the paper would be helpful. 

5) Finally, the authors claim state-of-the-art results. However, this is only
on a single MaxSimC metric. Other work has achieved overall better results
using the AvgSimC metric. So, while state-of-the-art isn't everything about a
paper, the claim that this paper achieves it - in the abstract and intro - is
at least a little misleading.

[Official Review · Reviewer 2 · rating 4 · confidence 4]
soundness 5 · originality 5 · clarity 4 · impact 3 · substance 3 · appropriateness 5 · meaningful comparison 3 · presentation format Poster

This paper describes a novel approach for learning multi-sense word
representations using reinforcement learning. A CBOW-like architecture is used
for sense selection, computing a score for each sense based on the dot product
between the sum of word embeddings in the current context and the corresponding
sense vector. A second module based on the skip-gram model is used to train
sense representations, given results from the sense selection module. In order
to train these two modules, the authors apply Q-Learning, where the Q-value is
provided by the CBOW-based sense selection module. The reward is given by the
skip-gram negative sampling likelihood. Additionally, the authors propose an
approach for determining the number of senses for each word non-parametrically,
by creating new senses when the Q-values for existing scores have a score under
0.5.

The resulting approach achieves good results under the "MaxSimC" metric, and
results comparable to previous approaches under "AvgSimC". The authors suggest
that their approach could be used to improve the performance for downstream
tasks by replacing word embeddings with their most probable sense embedding. It
would have been nice to see this claim explored, perhaps in a sequential
labeling task such as POS-tagging or NER, especially in light of previous work
questioning the usefulness of multi-sense representations in downstream tasks.
I found it somewhat misleading to suggest that relying on MaxSimC could reduce
overhead in a real world application, as the sense disambiguation step (with
associated parameters) would still be required, in addition to the sense
embeddings. A clustering-based approach using a weighted average of sense
representations would have similar overhead. The claims about improving over
word2vec using 1/100 of the data are also not particularly surprising on SCWS.
These are misleading contributions, as they do not advance/differ much from
previous work.

The modular quality of their approach results in a flexibility that I think
could have been explored further. The sense disambiguation module uses a vector
averaging (CBOW) approach. A positive aspect of their model is that they should
be able to substitute other context composition approaches (using alternative
neural architecture composition techniques) relatively easily.

The paper applies an interesting approach to a problem that has been explored
now in many ways. The results on standard benchmarks are comparable to previous
work, but not particularly surprising/interesting. However, the approach goes
beyond a simple extension of the skip-gram model for multi-sense representation
learning by providing a modular framework based on reinforcement learning.
Ideally, this aspect would be explored further. But overall, the approach
itself may be interesting enough on its own to be considered for acceptance, as
it could help move research in this area forward.

* There are a number of typos that should be addressed (line
190--representations*, 331--selects*, 492--3/4th*).

NOTE: Thank you to the authors for their response.

[Official Review · Reviewer 3 · rating 2 · confidence 4]
soundness 5 · originality 5 · clarity 3 · impact 3 · substance 4 · appropriateness 5 · meaningful comparison 3 · presentation format Poster

TMP
Strength: The paper propose DRL-Sense model that shows a marginal improvement
on SCWS dataset and a significant improvement on ESL-50 and RD-300 datasets.

Weakness:
The technical aspects of the paper raise several concerns:
Could the authors clarify two drawbacks in 3.2? The first drawback states that
optimizing equation (2) leads to the underestimation of the probability of
sense. As I understand, eq(2) is the expected reward of sense selection, z_{ik}
and z_{jl} are independent actions and there are only two actions to optimize.
This should be relatively easy. In NLP setting, optimizing the expected rewards
over a sequence of actions for episodic-task has been proven doable (Sequence
Level Training with Recurrent Neural Networks, Ranzato 2015) even in a more
challenging setting of machine translation where the number of actions ~30,000
and the average sequence length ~30 words. The DRL-Sense model has maximum 3
actions and it does not have sequential nature of RL. This makes it hard to
accept the claim about the first drawback.

The second drawback, accompanied with the detail math in Appendix A, states
that the update formula is to minimize the likelihood due to the log-likelihood
is negative. Note that most out-of-box optimizers (Adam, SGD, Adadelta, …)
minimize a function f, however, a common practice when we want to maximize f we
just minimize -f. Since the reward defined in the paper is negative, any
standard optimizer can be use on the expected of the negative reward, which is
always greater than 0. This is often done in many modeling tasks such as
language model, we minimize negative log-likelihood instead of maximizing the
likelihood. The authors also claim that when “the log-likelihood reaches 0,
it also indicates that the likelihood reaches infinity and computational flow
on U and V” (line 1046-1049). Why likelihood→infinity? Should it be
likelihood→1?

Could the authors also explain how DRL -Sense is based on Q-learning? The
horizon in the model is length of 1. There is no transition between
state-actions and there is not Markov-property as I see it (k, and l are draw
independently). I am having trouble to see the relation between Q-learning and
DRL-Sense.  In (Mnih et al., 2013), the reward is given from the environment
whereas in the paper, the rewards is computed by the model. What’s the reward
in DRL-Sense? Is it 0, for all the (state, action) pairs or the cross-entropy
in eq(4)?  

Cross entropy is defined as H(p, q) = -\sum_{x} q(x)\log q(x), which variable
do the authors sum over in (4)? I see that q(C_t, z_{i, k}) is a scalar
(computed in eq(3)), while Co(z_{ik}, z_{jl}) is a distribution over total
number of senses eq(1). These two categorial variables do not have the same
dimension, how is cross-entropy H in eq(4) is computed then?

Could the authors justify the dropout exploration? Why not epsilon-greedy
exploration? Dropout is often used for model regularization, preventing
overfitting. How do the authors know the gain in using dropout is because of
exploration but regularization?

The authors states that Q-value is a probabilistic estimation (line 419), can
you elaborate what is the set of variables the distribution is defined? When
you sum over that set of variable, do you get 1? I interpret that Q is a
distribution over senses per word, however  definition of q in eq(3) does not
contain a normalizing constant, so I do not see q is a valid distribution. This
also related to the value 0.5 in section 3.4 as a threshold for exploration.
Why 0.5 is chosen here where q is just an arbitrary number between (0, 1) and
the constrain \sum_z q(z) = 1 does not held? Does the authors allow the
creation of a new sense in the very beginning or after a few training epochs? I
would image that at the beginning of training, the model is unstable and
creating new senses might introduce noises to the model.  Could the authors
comment on that?

General discussion
What’s the justification for omitting negative samples in line 517? Negative
sampling has been use successfully in word2vec due to the nature of the task:
learning representation. Negative sampling, however does not work well when the
main interest is modeling a distribution p() over senses/words. Noise
contrastive estimation is often preferred when it comes to modeling a
distribution. The DRL-Sense, uses collocation likelihood to compute the reward,
I wonder how the approximation presented in the paper affects the learning of
the embeddings.

Would the authors consider task-specific evaluation for sense embeddings as
suggested in recent research [1,2]

[1] Evaluation methods for unsupervised word embeddings. Tobias Schnabel, Igor
Labutov, David Mimno and Thorsten Joachims.

[2] Problems With Evaluation of Word Embeddings Using Word Similarity Tasks .
Manaal Faruqui, Yulia Tsvetkov, Pushpendre Rastogi, Chris Dyer

---
I have read the response.